# Durvalumab Treatment Patterns for Patients with Unresectable Stage III Non-Small Cell Lung Cancer in the Veterans Health Administration (VHA): A Nationwide, Real-World Study

Amanda M. Moore [1,2], Zohra Nooruddin [2,3], Kelly R. Reveles [1,2,3], Paromita Datta [2,3], Jennifer M. Whitehead [2,3], Kathleen Franklin [2,3], Munaf Alkadimi [2,3], Madison H. Williams [4], Ryan A. Williams [4], Sarah Smith [2,3], Renee Reichelderfer [3], Ion Cotarla [5], Lance Brannman [6], Andrew Frankart [7], Tiernan Mulrooney [5], Kristin Hsieh [5,8], Daniel J. Simmons [5], Xavier Jones [1,3] and Christopher R. Frei [1,2,3,*]

1   College of Pharmacy, The University of Texas at Austin, Austin, TX 78712, USA; mandy.moore@utexas.edu (A.M.M.); revelesk@uthscsa.edu (K.R.R.); jonesx@uthscsa.edu (X.J.)
2   Long School of Medicine, University of Texas Health San Antonio, San Antonio, TX 78229, USA; nooruddinz@uthscsa.edu (Z.N.); paromita.datta@va.gov (P.D.); jennifer.aldridge8@gmail.com (J.M.W.); kathleen.franklin2@va.gov (K.F.); munafalkadhimi@hotmail.com (M.A.); smiths21@uthscsa.edu (S.S.)
3   Audie L. Murphy Veterans Hospital, South Texas Veterans Health Care System, San Antonio, TX 78229, USA; renee.reichelderfer@va.gov
4   MD Anderson Cancer Center, Houston, TX 77030, USA; mhwilliams@mdanderson.org (M.H.W.); rawilliams6@mdanderson.org (R.A.W.)
5   AstraZeneca US Medical Affairs, Gaithersburg, MD 20878, USA; ion.cotarla@astrazeneca.com (I.C.); tiernan.mulrooney@astrazeneca.com (T.M.); kristin.hsieh@astrazeneca.com (K.H.); daniel.simmons@astrazeneca.com (D.J.S.)
6   College of Pharmacy, University of Utah, Salt Lake City, UT 84112, USA; lance.brannman@utah.edu
7   Department of Radiation Oncology, University of Cincinnati, Cincinnati, OH 45267, USA; frankaaj@ucmail.uc.edu
8   Department of Radiation Oncology, Icahn School of Medicine at Mount Sinai, New York, NY 10029, USA
*   Correspondence: freic@uthscsa.edu; Tel.: +1-(210)-567-8371; Fax: +1-(210)-567-8328

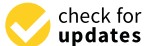



**Simple Summary:** Key Objective: Why do patients with unresectable stage III non-small cell lung cancer (NSCLC) experience durvalumab treatment initiation delays, interruptions, and discontinuations post-chemoradiotherapy (CRT)? Knowledge Generated: Toxicities are one of the main reasons for durvalumab treatment initiation delays, interruptions, and discontinuations. Relevance: Patients could benefit from improved strategies to prevent, identify, and manage CRT and durvalumab toxicities timely and effectively.

**Abstract:** Background: Durvalumab is approved for the treatment of adults with unresectable stage III non-small cell lung cancer (NSCLC) post-chemoradiotherapy (CRT). This real-world study describes patient characteristics and durvalumab treatment patterns (number of doses and therapy duration; treatment initiation delays, interruptions, discontinuations, and associated reasons) among VHA-treated patients. Methods: This was a retrospective cohort study of adults with unresectable stage III NSCLC receiving durvalumab at the VHA between 1 January 2017 and 30 June 2020. Patient characteristics and treatment patterns were presented descriptively. Results: A total of 935 patients were included (median age: 69 years; 95% males; 21% Blacks; 46% current smokers; 16% ECOG performance scores ≥ 2; 50% squamous histology). Durvalumab initiation was delayed in 39% of patients (*n* = 367). Among the 200 patients with recorded reasons, delays were mainly due to physician preference (20%) and CRT toxicity (11%). Overall, patients received a median (interquartile range) of 16 (7–24) doses of durvalumab over 9.0 (2.9–11.8) months. Treatment interruptions were experienced by 19% of patients (*n* = 180), with toxicity (7.8%) and social reasons (2.6%) being the most cited reasons. Early discontinuation occurred in 59% of patients (*n* = 551), largely due to disease progression (24.2%) and toxicity (18.2%). Conclusions: These real-world analyses corroborate PACIFIC study results in terms of the main reasons for treatment discontinuation in a VHA population with worse prognostic factors, including older age, predominantly male sex, and poorer performance score. One of the main reasons for durvalumab initiation delays, treatment interruptions, or discontinuations was due to

toxicities. Patients could benefit from improved strategies to prevent, identify, and manage CRT and durvalumab toxicities timely and effectively.

**Keywords:** lung cancer; pharmacoepidemiology; durvalumab; chemotherapy; radiotherapy; immunotherapy; chemoradiotherapy; stage III NSCLC

## 1. Introduction

Lung cancer is the most common cause of cancer-related death in the world with a death rate greater than breast, colon, and pancreatic cancers combined [1]. Non-small cell lung cancer (NSCLC) accounts for the majority of cases and is typically identified at a later stage, with one-third of patients having stage III, locally advanced disease, at the time of diagnosis [2,3].

Definitive platinum-based doublet chemoradiotherapy (CRT) has traditionally been used in patients with unresectable stage III NSCLC [4] with historical 5-year survival rates of 15 to 32%, with a median overall survival (OS) of 29 months or less [4,5]. The advent of immune checkpoint inhibitors in recent years for use as consolidation therapy has been shown to substantially improve survival outcomes in this population [6–10].

Durvalumab (IMFINZI®), an immune checkpoint inhibitor, is a selective, high-affinity human immunoglobulin G1 kappa monoclonal antibody that blocks programmed death-ligand 1 (PD-L1) binding to programmed cell death protein 1 (PD-1) and CD80, enabling T cells to recognize and kill tumor cells. Approval of durvalumab's use post-CRT in unresectable stage III NSCLC was based on the PACIFIC study, a randomized, placebo-controlled, phase III trial with durvalumab initiated 1 to 42 days post-CRT and dosed every two weeks. At the first interim analysis, durvalumab significantly prolonged progression-free survival (PFS) when compared to placebo (HR = 0.52; 95% CI, 0.42 to 0.65; $p < 0.001$; median, 16.8 vs. 5.6 months) [11] On the basis of these results, durvalumab was approved by the U.S. Food and Drug Administration (FDA) in February 2018 [12]. Durvalumab benefits were confirmed in a large observational study, PACIFIC-R, which included 1399 patients from 11 countries, who received durvalumab treatment through an ex-US early access program [13]. The latest analyses of the PACIFIC trial published in 2022 demonstrated that updated PFS (HR = 0.55; 95% CI, 0.45 to 0.68; median 16.9 vs. 5.6 months) and OS (HR = 0.72; 95% CI, 0.59 to 0.89; median 47.5 vs. 29.1 months) remained consistent with primary analyses. Estimated 5-year survival rates for durvalumab (43% OS and 33% PFS) established a new benchmark for success in patients with unresectable stage III NSCLC [14].

Despite robust clinical trial data, relatively limited evidence exists evaluating the real-world safety and effectiveness of durvalumab in clinical practice [15–20]. Durvalumab has been used extensively in the Veterans Health Administration (VHA) in patients with stage III NSCLC. A recent study published by investigators at the VHA showed that veterans treated with durvalumab consolidation had improved PFS and OS compared to historical controls, but OS was lower compared to the PACIFIC cohort [20]. Unfortunately, this study did not provide information on treatment initiation delays (TID), treatment interruptions (TI), and their associated reasons.

It is important to investigate real-world durvalumab treatment patterns, including rates and reasons for TID (defined as durvalumab initiation after 42 days post-CRT), TI, and treatment discontinuations (TD) to help inform future research and improve clinical practice and patient outcomes. The primary objective of this study was to describe patient characteristics and durvalumab treatment patterns in patients presenting to the VHA with unresectable stage III NSCLC following platinum-based CRT. The secondary objectives were to quantify and describe reasons for durvalumab TID, TI, and TD. This is one of the largest US national studies to date to evaluate important durvalumab real-world treatment patterns.

## 2. Materials and Methods

### 2.1. Data Source

The VHA has health care facilities in all 50 states, Washington, DC, USA, and surrounding territories, and maintains an electronic health record (EHR), which includes structured administrative, clinical, laboratory, and pharmacy data repositories. These repositories include data from both hospital and outpatient clinic settings. A more complete description of the VHA data is in Frei et al. [21] Four national Veterans Affairs (VA) databases, namely the VA Medical SAS® Datasets (both inpatient and outpatient), the VA Vital Status File, the VA Decision Support System Datasets, and the VA Assistant Deputy Under Secretary of Health Annual Enrollment Files, were linked using a unique patient identifier to develop an analytic dataset.

### 2.2. Study Design

This was a retrospective cohort study of patients with unresectable stage III NSCLC treated with durvalumab in the VHA system. This observational study required no intervention or interference with standard medical care and was approved by the University of Texas Health San Antonio Institutional Review Board and the South Texas Veterans Health Care System Research & Development Committee. Historical data (medical history) were examined for one year prior to the study period. The index date was defined as the date of initiation of durvalumab. Patients were followed until whichever occurred first: their last VHA visit, loss to follow-up, record of death, or the end of the follow-up period on 1 April 2021. Data were abstracted from structured data repositories to identify the initial patient cohort and several baseline patient characteristics [22]. Trained data abstractors manually collected additional information from medical charts, including durvalumab treatment patterns. Reasons for TID, TI, and TD were derived from physician notes. "Physician preference" meant that the physician decided to make the change. Similarly, "patient preference" meant that the patient decided to make the change.

### 2.3. Study Population

The study population consisted of adults with a diagnosis of unresectable stage III NSCLC who initiated durvalumab treatment in a VHA facility from 1 January 2017 to 30 June 2020. Patients were included in the study using criteria applied to the VHA EHR data and manual chart reviews. The EHR inclusion criteria were: (1) age 18 years or older; (2) inpatient or outpatient lung cancer diagnosis (ICD-10 codes C34X or D022X) between 1 January 2017, and 30 June 2020; and (3) an order for durvalumab (drug name, HCPCS C9492 or J9173, or NDC 0310-4500-12 or 0310-4611-50) between 1 January 2017, and 30 June 2020. The subsequent chart review criteria, used to confirm patient inclusion derived from the EHR data, were: (1) diagnosis of stage III NSCLC via a pathology report during the cohort inclusion period; (2) confirmation of unresectable tumor status in clinical notes; (3) receipt of CRT; and (4) initiation of durvalumab (within 1 to 42 days post-CRT and dosed every two weeks) during the cohort inclusion period. Patients were excluded from the study if they met the following EHR and chart review-derived criteria: (1) non-NSCLC histology; (2) non-stage III tumor classification; (3) resectable tumor status; (4) durvalumab receipt preceding the study inclusion period; (5) durvalumab not received during the study period; or (6) durvalumab therapy ongoing at the end of the study.

### 2.4. Data and Statistical Analysis

Patient demographics, clinical characteristics, and durvalumab treatment patterns were analyzed descriptively. Patients were categorized as experiencing durvalumab TID if the time from CRT completion to durvalumab initiation was greater than 42 days (the maximum initiation time defined in the PACIFIC study) [11]. Given durvalumab was dosed every two weeks, TI was defined as greater than 28 days between durvalumab infusions (missing > 2 infusions). TD was defined if more than 28 days passed from the last durvalumab dose with no new durvalumab restart. A corrected duration of therapy

(DOT) was calculated as DOT minus the days contributed by TI. Reasons for TID were calculated based on the patient subpopulation with recorded reasons in the chart ($n = 200$), while reported reasons for TI and TD were calculated as a proportion of the entire study population ($n = 935$), to match methodology used in the PACIFIC trial.

## 3. Results

Overall, 1185 patients met the EHR inclusion criteria and 250 patients were excluded during chart review due to: non-NSCLC histology ($n = 46$); non-stage III classification ($n = 162$); resectable tumor status ($n = 81$); durvalumab not received by patient ($n = 56$); and durvalumab therapy ongoing at end of study ($n = 43$) (exclusion categories were not mutually exclusive). A total of 935 patients were included in the study.

### 3.1. Patient Characteristics

Baseline demographics and clinical characteristics, including prior CRT characteristics, are listed in Table 1. Patients had a median age of 69 years (IQR, 65–72), with 76% aged 65 years or older. Patients were predominately male (95%), White (78%), and current or former smokers (97%). The median (IQR) Charlson age-adjusted comorbidity index was 6 (5–7) and the most frequently observed comorbidities at baseline included: chronic obstructive pulmonary disease (COPD) (70%), diabetes (33%), peripheral vascular disease (23%), cerebrovascular disease (14%), congestive heart failure (13%), renal disease (12%), and chronic liver disease (11%). Most patients had Eastern Cooperative Oncology Group (ECOG) scores of 0–1 (64%), 16% had a score of 2 or greater, and 20% had missing or unknown ECOG scores. Approximately half of the patients had NSCLC squamous cell subtype on histology (50%). Tumor expression of PD-L1 was available for 340 patients (37%) with 65% of these patients having PD-L1 expression ≥1% and 35% with negative/negligible expression (<1%).

**Table 1.** Baseline demographic and clinical characteristics.

| Characteristic | All (*n* = 935) |
|---|---|
| Age (years), median (IQR) | 69 (65–72) |
| Age groups, *n* (%) | -- |
| <65 years | 229 (24) |
| 65 to 74 years | 567 (61) |
| >74 years | 139 (15) |
| Male, *n* (%) | 891 (95) |
| Race, *n* (%) | -- |
| Missing/unknown | 1 (<1) |
| White | 726 (78) |
| Black | 198 (21) |
| Other | 10 (1) |
| Charlson score, median (IQR) | 3 (3–5) |
| Charlson Age score, median (IQR) | 6 (5–7) |
| Selected comorbidities, *n* (%) | -- |
| Congestive heart failure | 123 (13) |
| COPD | 659 (70) |
| Cerebrovascular disease | 131 (14) |
| Dementia | 15 (2) |
| Diabetes | 306 (33) |

**Table 1.** *Cont.*

| Characteristic | All (*n* = 935) |
|---|---|
| Hemi/paraplegia | 9 (1) |
| HIV/AIDS | 5 (1) |
| Liver disease | 107 (11) |
| Myocardial infarction | 65 (7) |
| Peptic ulcer disease | 17 (2) |
| Peripheral vascular disease | 219 (23) |
| Renal disease | 109 (12) |
| ECOG performance status score, *n* (%) | -- |
| Missing/unknown | 181 (20) |
| 0–1 | 602 (64) |
| 2–3 | 152 (16) |
| NSCLC histologic subtype, *n* (%) | -- |
| Missing/unknown | 56 (6) |
| Squamous cell | 469 (50) |
| Non-squamous cell | 400 (43) |
| Mixed | 10 (1) |
| PD-L1 tumor expression level, *n* (%) | -- |
| Missing/unknown | 595 (63) |
| Available PD-L1 test results | 340 (37) |
| PD-L1 expression level <1% among those with results | 118 (35) |
| PD-L1 expression level ≥1% among those with results | 222 (65) |
| Prior chemotherapy, *n* (%) | -- |
| Cisplatin-based chemotherapy | 106 (11) |
| Carboplatin-based chemotherapy | 817 (88) |
| Other | 12 (1) |
| Total chemotherapy weeks, median (IQR) | 6 (5–7) |
| Total radiation dose (Gy), *n* (%) | -- |
| Missing/unknown | 99 (10) |
| <54 | 27 (3) |
| 54–66 | 728 (78) |
| 67–74 | 74 (8) |
| >74 | 7 (1) |
| Radiation fractions, median (IQR) | 30 (30–33) |
| CRT type, *n* (%) | -- |
| Sequential | 15 (2) |
| Concurrent | 920 (98) |
| Time from end of CRT to first scan (days), median (IQR) | 30 (20–43) |
| Time from end of CRT to first scan, *n* (%) | -- |
| Missing/unknown | 128 (14) |
| <42 days | 590 (63) |
| ≥42 days | 217 (23) |

**Table 1.** *Cont.*

| Characteristic | All (*n* = 935) |
|---|---|
| CRT response [a], *n* (%) | -- |
|     Missing/unknown | 144 (15) |
|     Complete response | 31 (3) |
|     Partial response | 623 (67) |
|     Stable disease | 98 (11) |
|     Progressive disease | 27 (3) |
|     Non-evaluable | 12 (1) |

IQR = interquartile range, COPD = chronic obstructive pulmonary disease, HIV = human immunodeficiency virus, AIDS = acquired immunodeficiency syndrome, NSCLC = non-small cell lung cancer, ECOG = Eastern Cooperative Oncology Group, PD-L1 = programmed death-ligand 1, CRT = chemoradiotherapy. [a] CRT response for patients with scan documented between 2 weeks before CRT end and up to 2 weeks after durvalumab initiation.

Nearly all patients (98%) had concurrent CRT. The majority of patients received carboplatin-based chemotherapy (88%). Patients received a median (IQR) of 6 (5–7) weeks of chemotherapy. A median (IQR) of 30 (30–33) radiation fractions were administered and 78% of patients received a total radiation dose between 54–66 Gy. Post-CRT imaging was performed a median (IQR) of 30 (20–43) days from end of CRT to first scan and 63% of patients received a scan within 6 weeks of CRT completion. Most patients (67%) had a partial response following CRT (Table 1).

*3.2. Durvalumab Treatment Patterns*

Approximately one-third of patients started durvalumab therapy in 2018 (34%), 48% in 2019, and 17% in 2020 (the patient inclusion period ended June 30, 2020). Durvalumab treatment was initiated with a delay in 39% of patients (*n* = 367). Among patients who experienced a TID, median (IQR) time from the end of CRT to initiation of durvalumab was 61 (49–80) days. Of the 200 patients with reported reasons in the chart, 20% experienced a TID due to physician preference, 11% due to toxicity, and 10.5% due to patient preference; the other reported reasons included decline in performance status (10%), system issues (9.5%), and social reasons (9%) (Figure 1).

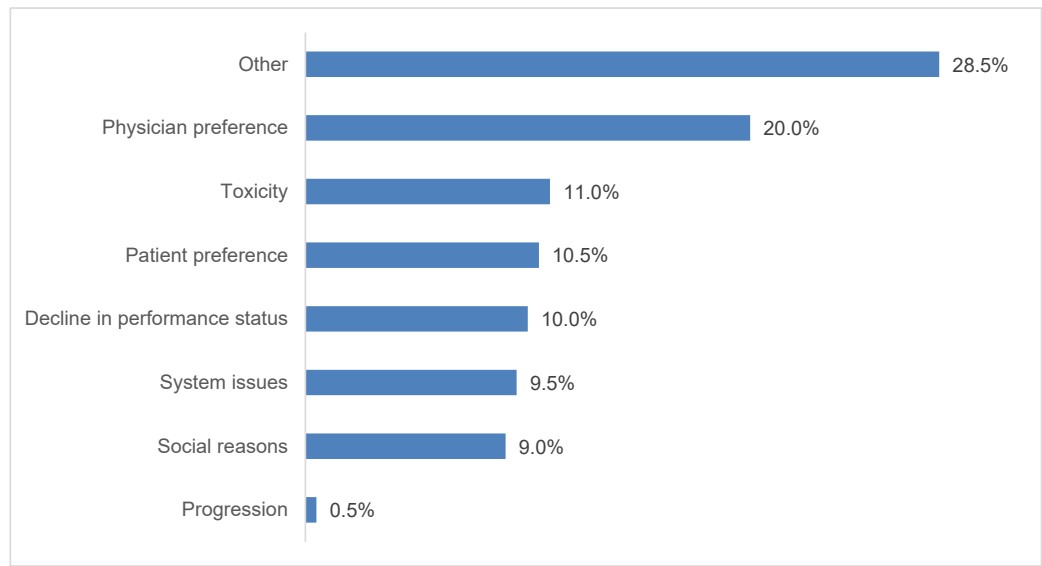

**Figure 1.** Reported reasons associated with durvalumab treatment initiation delays, *n* = 200. Additional categories with zero patients reported: cost-related and insurance-related. Treatment initiation delays were identified for 386 patients, but reasons were recorded for only 200 patients.

Overall, patients received a median (IQR) of 16 (7–24) durvalumab infusions (dosed every two weeks) over a median (IQR) DOT of 9 (2.9–11.8) months. Interruptions in durvalumab therapy were reported in 19% of patients (*n* = 180) and the median (IQR) duration of TI was 53 (39–90) days. After accounting for the number of days contributed by TI, the corrected median (IQR) DOT was 8.4 (2.8–11.7) months (Table 2). Toxicity was the most commonly reported reason for TI (7.8% of the entire study population), followed by social reasons (2.6%), patient preference (1.4%), physician preference (1.2%) and decline in performance status (1%) (Figure 2).

**Table 2.** Durvalumab therapy.

| Characteristic | All (*n* = 935) |
| --- | --- |
| Time to durvalumab initiation (days), median (IQR) | 39 (28–54) |
| Patients with durvalumab initiation delays [a], *n* (%) | 367 (39) |
| Duration of treatment initiation delay (days), median (IQR) | 61 (49–80) |
| Durvalumab duration of therapy (months), median (IQR) | 9.0 (2.9–11.8) |
| Durvalumab total doses/infusions, median (IQR) | 16 (7–24) |
| Patients with durvalumab interruptions [b], *n* (%) | 180 (19) |
| Number of durvalumab interruptions, median (IQR) | 1 (1–1) |
| Duration of durvalumab interruptions (days), median (IQR) | 53 (39–90) |
| Durvalumab corrected duration of therapy (months) [c], median (IQR) | 8.4 (2.8–11.7) |
| Durvalumab treatment discontinuations, *n* (%) | 551 (59) |
| Completed planned treatment, *n* (%) | 384 (41) |

IQR = interquartile range. [a] Durvalumab treatment delay defined as more than 42 days from end of CRT to initiation of durvalumab. [b] Durvalumab treatment interruptions defined as more than 28 days between durvalumab infusions. [c] Durvalumab corrected duration of therapy defined as duration of therapy minus days contributed by treatment interruptions.

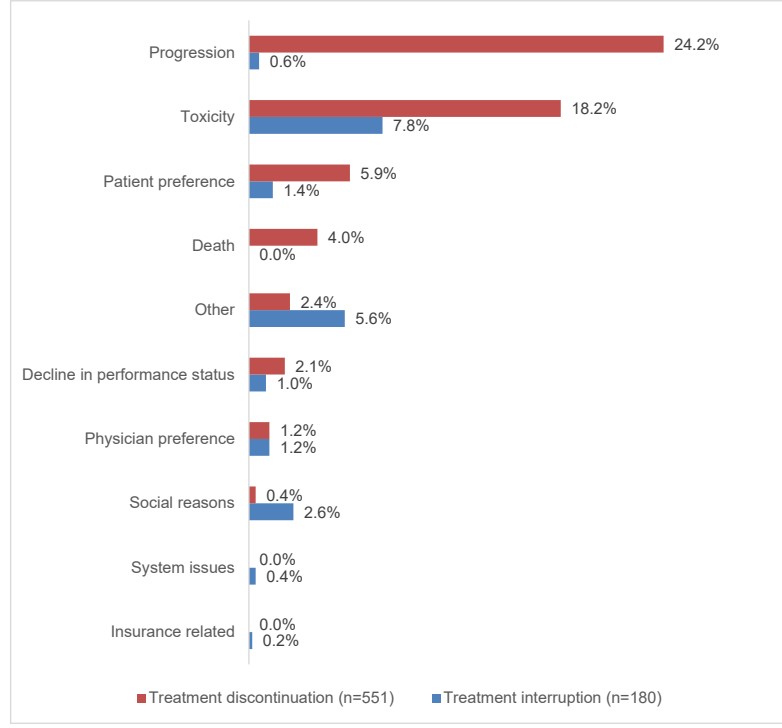

**Figure 2.** Reported reasons associated with durvalumab treatment interruptions and discontinuations. Multiple reasons could be reported for treatment interruptions and discontinuations.

Forty-one percent of patients (*n* = 384) completed the planned durvalumab treatment and 59% (*n* = 551) discontinued treatment early (Table 2). Of the entire study population, 24.2% of patients discontinued due to progression and 18.2% due to toxicity (18.2%); other reasons included patient preference (5.9%), death (4%), decline in performance status (2.1%) and physician preference (1.2%).

Patients with ECOG scores of 0–1 vs. 2–3 experienced similar TID, TI, and TD. Finally, time to durvalumab initiation declined by nearly 20 days from 2017 to 2020, possibly as a result of physicians becoming more accustomed to durvalumab use.

## 4. Discussion

This study aimed to describe patient characteristics and durvalumab treatment patterns in patients presenting to the VHA with unresectable stage III NSCLC. With a sample size of 935 patients, this is one of the largest national studies to date to examine the real-world clinical use of durvalumab, reporting the frequency of and reasons for TID, TI, and TD. The duration of follow-up in this study was 9 months, compared to 7 months in a prior real-world study [20]. Nearly two-in-five patients had delayed initiation of durvalumab therapy. About one-in-five patients experienced an interruption in treatment and more than half of the patients permanently discontinued durvalumab early.

Patients in this study were selected based on real-world durvalumab use as prescribed by treating physicians. In a departure from the PACIFIC trial criteria, our study included patients with significant comorbidities, poor performance status, treatment initiation more than 42 days beyond the end of CRT, or progressive disease on post-CRT imaging. Compared to the PACIFIC durvalumab cohort, patients in this study were older (median age of 69 years vs. 64 years in PACIFIC), more likely to be male (95% vs. 70% in PACIFIC), and more likely to be current smokers (46% vs. 17% in PACIFIC). Our patient cohort included more Black patients (21% vs. 3% in PACIFIC) and relatively fewer Asian patients (~1% vs. 25% in PACIFIC).

Tumor characteristics and CRT parameters were largely similar in this study with those in the durvalumab arm of the PACIFIC trial. In both studies, roughly half of the patients had squamous cell histology (50% vs. 47% in PACIFIC), almost all patients had concurrent CRT (98% vs. ~100% in PACIFIC). Relatively fewer patients had radiation doses ranging from 54–66 Gy (78% vs. 93% in PACIFIC). More patients in our study had a partial response from CRT as recorded by treating physicians (67% vs. 49% in PACIFIC) and received a carboplatin-based chemotherapy regimen (88% vs. 42% in PACIFIC) rather than a cisplatin-based regimen (12% vs. 58% in PACIFIC) [11]. Patients treated in VHA sites are likely to have a higher incidence of kidney disease than other populations and therefore more likely to be treated with carboplatin [23].

As part of the rationale for the PACIFIC trial, preclinical evidence has suggested that chemotherapy and radiotherapy might upregulate PD-L1 expression on tumor cells and prime T cells, thus it was hypothesized that durvalumab initiation should not be delayed after CRT to provide maximal clinical benefit [11,24–27]. Effects of treatment initiation delays on outcomes are being investigated [14,28–30]. This study and other real-world studies on durvalumab have shown that patients commonly experience delays in treatment initiation beyond the 42 days after CRT as specified in the PACIFIC trial [16,19,31]. In our study, 39% of patients experienced a delay in initiation of durvalumab therapy; in contrast, 57% of patients experienced delays in a U.S. community oncology network [31], and 78% of patients had delays in a study of Italian Centers in the Association of Radiotherapy and Clinical Oncology thoracic oncology network [16]. Our study reports a notable improvement in the proportion of durvalumab TID compared to these other groups, which may be representative of improved access to care in an equal-access health system such as the VHA. This improvement is especially remarkable considering our patient cohort included those with older age, significant comorbidities, and/or poor performance status, all of which may relate to social reasons that may affect time to initiation of treatment. Specifically, such patients may have increased difficulty accessing care, including scheduling and coordi-

nating their appointments and travel to/from such appointments and actually traveling to/from appointments, resulting in a delay of durvalumab treatment initiation. Physician preference and toxicity from CRT represented the most commonly cited reasons for TID as derived from patient charts. The extended time between the end of CRT and start of durvalumab may represent recovery time from CRT adverse events, as well as logistical delays in treatment planning in a real-world setting [29]. Notably, the median time from end of CRT to first scan was 30 days, with 23% of patients receiving their first post-CRT imaging more than 42 days from the end of CRT. This may account for physician preference, as the lack of post-CRT scans limits treatment decision-making. Due to the aforementioned social barriers and other patient-related barriers that may limit care, physicians must weigh the benefits and challenges of a timely first post-CRT imaging and initiation of post-CRT treatment and discuss with the patients to determine the optimal management.

Durvalumab was initially approved to be administered every two weeks. In November 2020, durvalumab was also approved to be administered every four weeks. Given the cutoff of our study is June 2020, all of our patients received durvalumab dosed every two weeks. Patients in this study received a median of 16 durvalumab infusions (dosed every two weeks) over a median of 9 months. This real-world patient population had less time in therapy when compared to the PACIFIC cohort (median 20 doses over 9.2 months) [32]; however, other real-world studies have reported shorter durations of therapy and fewer doses when compared to our study, although some of these studies did not exclude patients with ongoing therapy which may skew results [16,20,33–35]. The results of our study reflect the resilience of the veteran population and highlight healthcare best practices at the VHA in a population that was older, with more comorbidities, and a meaningful proportion of ECOG performance status scores of 2 or greater.

Approximately one-fifth of patients (19%) experienced interruptions in treatment. Most patients experienced a single TI of about two months (median 53 days), with the most commonly reported reason for TI being toxicity. Durvalumab therapy was discontinued prematurely in over half of patients (59%), most commonly due to disease progression and toxicity. In terms of percentage of patients having treatment discontinuation due to adverse effects, the PACIFIC trial reported 15.4% compared to 18.2% in our study. The higher treatment discontinuation rate due to adverse effects in our study might be due to multiple worse prognostic factors at baseline in our patient cohort, such as older age, poorer performance score, and more comorbidities. Further research to identify which prognostic factors contribute to adverse effects that result in treatment discontinuation is warranted. These discontinuation rates and reasons have been observed in other populations, with up to 57% of patients discontinuing treatment for primarily disease progression and toxicities in other analyses [16,20,33,36,37]. The results of our analysis substantiate findings reported in the PACIFIC trial, in which approximately 50% of patients discontinued durvalumab early, also due to disease progression and adverse events, despite this being a real-world study including patients with poorer ECOG performance status and more comorbidities at baseline [11].

Durvalumab was approved by the U.S. FDA given it significantly prolonged PFS and OS in unresectable stage III NSCLC post-CRT when compared to placebo [11,12]. This durable benefit in PFS and OS was redemonstrated in the five-year analysis of the PACIFIC trial [14]. Although a significant portion of patients discontinued treatment due to disease progression or toxicities, the effectiveness and safety of the drug in the remaining patients are notable. To date, durvalumab is the only approved immunotherapy agent for unresectable stage III NSCLC post-CRT. Multiple phase III clinical trials utilizing other immunotherapy agents are being compared to the PACIFIC regimen, which is currently the standard of care in this setting.

This study has several strengths. First, we collected data on all patients treated with durvalumab for unresectable stage III NSCLC at any VHA facility over a 42-month period. The comprehensive VHA data enabled us to study many relevant clinical variables. We used a combination of electronic data extraction from national datasets and manual chart

review by trained data abstractors to validate data and minimize risk of misclassification bias. This is one of the largest national studies to date to evaluate important real-world durvalumab treatment patterns, and the first to report associated reasons for TID and TI, giving valuable insight into the real-world clinical use of durvalumab in NSCLC.

This study also has potential limitations. The VHA is a predominantly older, White, male population and study findings might not be generalizable to non-VHA settings. EHR data are created for the purpose of patient care, not research, and might contain errors. There were variations in the extent to which patients' comorbidities and reasons for durvalumab TID, TI, and TD were reported; reasons were tabulated if they occurred prior to TID, TI, or TD, but causality was not always explicitly mentioned in the patient charts and may be subject to documentation bias. Durvalumab treatment delay was common and could be due to a delay in disease re-staging or a decline in performance status. Unfortunately, information on restaging when treatment was delayed is not available. Short follow-up times in some patients might impact the treatment durations, although to mitigate this bias, patients were excluded if durvalumab therapy was ongoing at the end of the study period. Given the retrospective design, the data might be subject to misclassification bias and confounding due to the use of administrative coding from the EHR for data collection. To mitigate these biases, manual data abstraction was used to validate key variables included in the study. Furthermore, we did not capture PFS, OS, radiotherapy technique and target area, or post-progression therapy in this study, so we are not able to assess those outcomes or provide a description. Additionally, we did not capture the timing of additional imaging, tumor biomarker testing, and follow-ups. We acknowledge these as limitations of this study and opportunities for future research. Finally, because this was a retrospective cohort study, with manual chart abstraction, we were often not able to determine the severity of the toxicity, or the seriousness of the adverse effects associated with the toxicity, due to lack of details recorded in the notes.

## 5. Conclusions

This is one of the largest US national studies to date to evaluate durvalumab therapy in unresectable stage III NSCLC. This study presented patterns of real-world clinical use of durvalumab and notably highlights the resilience of the VHA patient population, which received a median duration of 9 months of therapy despite a meaningful proportion of patients having an ECOG performance status of 2 or greater. Nearly two in five patients experienced delays in treatment initiation, about one in five patients experienced treatment interruptions and over half of patients permanently discontinued treatment early. The results of this study suggest that improved coordination of timely imaging after the end of CRT may prevent treatment initiation delays. Additionally, early recognition and proper management of treatment toxicities may prevent prolonged interruptions or premature discontinuations of durvalumab therapy. Altogether, this analysis corroborates the PACIFIC trial findings with durvalumab in a real-world setting, despite an older, less fit population, with more comorbidities and worse prognostic factors.

**Author Contributions:** Study concept and design: A.M.M., Z.N., K.R.R., L.B., I.C. and C.R.F. Statistical analysis: A.M.M. and C.R.F. Interpretation of data: A.M.M., Z.N., K.R.R. and C.R.F. Drafting of the manuscript: A.M.M. and C.R.F. Critical revision of the manuscript for important intellectual content: all authors (A.M.M., Z.N., K.R.R., P.D., J.M.W., K.F., M.A., M.H.W., R.A.W., S.S., R.R., I.C., L.B., A.F., T.M., K.H., D.J.S., X.J. and C.R.F.). Study supervision: Z.N. and C.R.F. All authors have read and agreed to the published version of the manuscript.

**Funding:** This research was funded by AstraZeneca. The APC was funded by AstraZeneca.

**Institutional Review Board Statement:** The study was conducted in accordance with the Declaration of Helsinki and approved by the Institutional Review Board (or Ethics Committee) of the University of Texas Health Science Center at San Antonio (protocol code HSC20200701H, approved 22 September 2020).

**Informed Consent Statement:** Patient consent was waived due to minimal risk to the study subjects, because the information collected was limited only to information that was recorded in records for purposes other than for this research study. Only the minimum information necessary to complete the research was recorded. The research involved existing data only. No patients or providers were contacted. The patient identifiers were maintained in limited-access directories on VA research servers behind the VA firewall at all times.

**Data Availability Statement:** The data are not publicly available due to patient privacy restrictions by the United States Department of Veterans Affairs.

**Acknowledgments:** Funding for the study was provided by AstraZeneca as a research grant to the Foundation for Advancing Veterans' Health Research (FAVHR), a non-profit entity within the Audie L. Murphy Veterans Hospital, San Antonio, TX, USA. The views expressed in this article are those of the authors and do not necessarily represent the views of the Department of Veterans Affairs or the authors' other affiliated institutions.

**Conflicts of Interest:** Z.N., K.R.R., P.D. and C.R.F. have received research grants (paid to FAVHR) from AstraZeneca in the last three years. L.B., I.C., T.M. and D.J.S. are employees of AstraZeneca. A.F. and K.H. were supported by the ASTRO-AstraZeneca Radiation Oncology Research Training Fellowship. All other authors declare no conflicts of interest.

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
