# Peer review of "Durvalumab Treatment Patterns for Patients with Unresectable Stage III Non-Small Cell Lung Cancer in the Veterans Health Administration (VHA): A Nationwide, Real-World Study"

_curroncol, doi:10.3390/curroncol30090611_

Round 1
Reviewer 1 Report
Interesting and well written paper.
The large series makes the manuscript of considerable prestige
In the first part of the discussion, the comparison between the series and the pacific trial is described. This part should be inserted in the results session ina dedicated paragraph.
Author Response
We want to thank the reviewers for their helpful comments and suggestions. We are happy to have been able to address these. The manuscript is better for having done so. Thank you for your help.

Reviewer 2 Report
This is an interesting manuscript showing real-world evidence from the use of durvalumab, a PD-L1 inhibitor, used for the treatment of Non-Small Cell Lung Cancer (NSCLC). The authors used information from a US based large registry for almost 3 years (2017-2020). The aim of the study is focusing on disease free survival analysis and discontinuation rates and adverse events of the treatment. All information were gathered retrospectively.
The study describes well the studied population and their commorbidities and lung cancer type and extent.
Only a few comments should be taken into consideration:
- The tables might be shorter.
- The authors refer to time to initiation of treatment. Please explain how physician preference and social reasons can affect that.
- Almost 1/3 of your study population has delay starting treatment >1 month. This could reflect on staging or performance. Please specify if restaging was performed when treatment was delayed.
- There is a superficial reference about "toxicity". Since the reported toxicity percentage leading to discontinuation you have to mention what kind of toxicity and if there were SAEs due to toxicity (hospitalization, death). This is crucial because it is about drug safety. A table could be added to clarify this point.
General comment
The accumulative percentage of disease progression and toxicity that lead to treratment discontinuation is 42,4%, which is quite high and puts the drug safety and effectiveness in question. Therefore, there should be a paragraph in discussion explaining if durvalumab is safe and effective, and perhaps compare it with similar drugs.
Author Response

(The authors gave the same response as above.)

Reviewer 3 Report
The manuscript entitled “Durvalumab treatment patterns for patients with unresectable stage III non-small cell lung cancer in the Veterans Health Administration (VHA): a nationwide, real-world study” describes the real-world data of durvalumab consolidation after chemoradiotherapy (CRT) among VHA treated patients.
As stated in the discussion section, the current study is one of the largest in the world.
However, the content remains a mere description of the administration pattern of durvalumab including the time to durvalumab initiation, duration and interruptions of durvalumab consolidation, as well as other background characteristics of patients such as chemotherapy regimens.
I think further analysis would make the manuscript valuable.
Therefore, I recommend the authors a major revision.
Major:
1) This study should also evaluate outcomes such as progression-free survival (PFS) and overall survival (OS).
2) The relationship between durvalumab consolidation duration and PFS or OS, as well as the impact of durvalumab discontinuation due to adverse events on PFS/OS should be included in the analysis.
3) Since the patients with performance status (PS) of 2-3 are included in this study (16%), which is different from the PACIFIC trial, the impact of poor PS on the outcomes should be also evaluated.
4) The time to durvalumab initiation after CRT may have changed as the treating physicians get accustomed to durvalumab use. Then, the relationship between the time to durvalumab initiation and the year when durvalumab was started would be of interest.
Minor:
1) What is the physician’s preference?
2) What dose the “VA priority” stand for?
Author Response
We want to thank the reviewers for their helpful comments and suggestions. We are happy to have been able to address these. The manuscript is better for having done so. Thank you for your help, including the great suggestion to look and see if durvalumab start year was associated with durvalumab time to treatment initiation. Thank you!

Reviewer 4 Report
General comment:
This real-world study describes patient characteristics and durvalumab treatment patterns among patients treated at the VHA,focusing on delays in durvalumab initiation, interruptions, discontinuations, and associated reasons. This article is well-written and analyzes the main reasons for durvalumab initiation delays, treatment interruptions or terminations in the real world. Therefore, I would like to give a " accept".
Introduction
The introduction of this article is well-written, providing a brief overview of NSCLC and the clinical trials related to durvalumab.
Materials and Methods
1. On page 3, line 129, it says that this study excluded patients who were still on durvalumab therapy at the end of the study, please explain the reason for this exclusion.
2. The article does not state when and how the patients were followed up, and whether imaging or tumor biomarkers were performed at a set time?
Results
1. The results do not present efficacy and safety data for durvalumab. It would have been better to show the PFS, OS and adverse effects of the patients.
2. Figure 2 shows that one of the reasons patients interrupted and discontinued durvalumab therapy was toxicity, and it would have been better to list specific adverse effects.
3. The article does not describe the patient's radiotherapy technique, or the determination of the radiotherapy target area. It would have been better if the description had been made.
4. Interruptions and discontinuations of treatment in some patients is due to disease progression, and it is recommended that post-progression therapeutic measures be demonstrated
Discussion
The discussion in this paper is well written and compares the results of this study with the results of the PACFIC study and other real world studies and discusses them.
Conclusion
1. Overall, the conclusion section is clearly written.
2. On page 10, line 315, it says that the results of this study suggest that improved coordination of timely imaging at the end of CRT can prevent delays in treatment initiation. It would be better if this was explained in the discussion section.
Author Response

(The authors gave the same response as above.)

Round 2
Reviewer 2 Report
The comments have been considered, and I hope they have helped the authors to improve this noteworthy manuscript.
Reviewer 3 Report
I understand the limitation of this study. I think that the manuscript was well revised.